The CLEC3B inhibits cellular proliferation and metastasis of cholangiocarcinoma through Wnt/β-catenin pathway

Wu Shengze 1
Wang Guangfeng 1
Xie Yuwei 1
Wu Tingting 2
Du Fangqian 1
Jin Chen 2
Dong Bingzi dongbingzi@hotmail.com 3 4
Zhu Chengzhan zhuchengz@qduhospital.cn 1 4
1 Department of Hepatobiliary and Pancreatic Surgery, The Affiliated Hospital of Qingdao University , Qingdao , Shandong , China
2 Department of Pediatric Surgery, The Affiliated Hospital of Qingdao University , Qingdao , Shandong , China
3 Department of Endocrinology and Metabolism, The Affiliated Hospital of Qingdao University , Qingdao , Shandong , China
4 Shandong Key Laboratory of Digital Medicine and Computer Assisted Surgery, The Affiliated Hospital of Qingdao University , Qingdao , Shandong , China
Haraguchi Tokuko
Electronic publication date: 2024 Nov 13
Publication date: 2024
Volume: 12
Electronic Location ID: e18497
Received 2024 Jun 27; Accepted 2024 Oct 18
Copyright: ©2024 Wu et al.
Copyright year: 2024
Copyright holder: Wu et al.
License: This is an open access article distributed under the terms of the Creative Commons Attribution License, which permits unrestricted use, distribution, reproduction and adaptation in any medium and for any purpose provided that it is properly attributed. For attribution, the original author(s), title, publication source (PeerJ) and either DOI or URL of the article must be cited.
License URL: https://creativecommons.org/licenses/by/4.0/

Keywords: CLEC3B, Cholangiocarcinoma (CCA), Proliferation, Migration, Calcium

Funding: Taishan Scholars Program of Shandong Province 2019010668 tsqn202312382 Natural Science Foundation of Shandong Province ZR2021MH171 ZR2023MH243 Shandong Higher Education Young Science and Technology Support Program 2020KJL005 2023KJ224 The study was supported by the Taishan Scholars Program of Shandong Province (grant number 2019010668 and NO. tsqn202312382), the Natural Science Foundation of Shandong Province (grant number ZR2021MH171, ZR2023MH243), and the Shandong Higher Education Young Science and Technology Support Program (grant number 2020KJL005 and 2023KJ224). The funders had no role in study design, data collection and analysis, decision to publish, or preparation of the manuscript.

==============================
Purpose

Cholangiocarcinoma (CCA) is a cancer of the biliary system, including intrahepatic and extrahepatic cholangiocarcinoma, and is highly aggressive. C-type lectins family member 3b (CLEC3B) is a Ca2+ binding transmembrane protein with different biological functions in a variety of cancers. The objective of this study was to explore the biological function of CLEC3B in CCA.

Methods

The CLEC3B gene was identified using the TCGA database and survival analysis of the cholangiocarcinoma clinical cohort. The expression CLEC3B cholangiocarcinoma and correlation with prognosis was investigated in our patient cohort. The effects of CLEC3B on proliferation, apoptosis, migration and invasion were verified in QBC939 and HUCCT1 cells. The effect of CLEC3B on the tumor formation was proved by xenograft tumor model in nude mice. The signaling pathway of CLEC3B in regulating CCA progression was further analyzed RNA sequencing and western blot.

Results

CLEC3B was decreased in the cholangiocarcinoma in the database. The mRNA and protein expression level of CLEC3B were significantly lower and correlated with poor overall survival in cholangiocarcinoma of our patient cohort. In vitro experiments proved that overexpression of CLEC3B can inhibit proliferation, migration and invasion in bile duct cancer cells. The CLEC3B was correlated with epithelial–mesenchymal transition and apoptosis. The calcium could promote the biological function of CLEC3B. The vivo study indicated that CLEC3B inhibited tumor formation. RNA sequencing indicating CLEC3B may transduce signal through e Wnt/β-catenin signaling pathway.

Conclusions

The CLEC3B inhibits cellular proliferation and migration of cholangiocarcinoma through the Wnt/β-catenin pathway.

Introduction

Cholangiocarcinoma, as a highly heterogeneous biliary malignancy, originates from biliary epithelial cells. It can occur anywhere in the biliary tract system. The incidence of cholangiocarcinoma is currently on the rise worldwide, accounting for about 15% of primary liver cancers and 3% of gastrointestinal malignancies (Banales et al., 2020). Surgical resection is the main treatment method for cholangiocarcinoma while the resection rate is low. The postoperative relapse rate is high and the response to chemotherapy is poor. Therefore, it is very important to deeply study the progression and metastasis mechanism of cholangiocarcinoma and seek breakthroughs in new therapeutic targets.

Type C lectin is the largest and most diverse family of mammalian carbohydrates-binding proteins, which share a common protein folding that allows for a uniform system of Ca2+ mediated carbohydrate recognition (Keller & Rademacher, 2020). Structure related Ca2+ dependency carbohydrate recognition domain structure is its mediated interaction between pathogen recognition and cells (Weis, Taylor & Drickamer, 1998). Cell–cell adhesion and immune response are both functions of C-type lectin domain proteins (Cambi & Figdor, 2009; Drickamer, 1999). There are total of 17 subgroups of proteins (Zelensky & Gready, 2005). Several members have been found to be related with tumor microenvironment and response to immunotherapy. A pan-cancer analysis has indicated that CLEC5A could predict cancer immunity and prognosis (Chen et al., 2022). In lung cancer and melanoma, CLEC2C/CD69 can be used to predict response to PD-1/PD-L1 blocking cancer immunotherapy (Hu et al., 2022). Bispecific CLEC9A-PD-L1 targeting type I interferons can be used to reshape the tumor microenvironment to an antitumor state (Van Lint et al., 2023). The biological function of Type C lectin needs to be further investigated.

CLEC3B/Tetranectin, a member of C-type lectin, is a Ca2+-binding protein. CLEC3B has been reported to be downregulated in several kinds of cancer (Liu et al., 2018; Xie, Jiang & Li, 2020). Low expression of CLEC3B is associated with low survival of HCC, and is negatively associated with immune cell infiltration and multiple immune biomarkers (Xie, Jiang & Li, 2020). CLEC3B was also downregulated in exosome of HCC. Through AMPK and VEGF signaling pathways, exosome CLEC3B can inhibit tumor metastasis and angiogenesis (Dai et al., 2019). The immunoinfiltration and immunoactivation of lung cancer may also be related to the significant downregulation of CLEC3B (Sun et al., 2020). However, CLEC3B, co-expressed with a-SMA in cancer-associated fibroblasts in CRC, was indicated to be a tumor progressor. The proliferation of colon cancer cells can be promoted by CLEC3B (Zhu et al., 2019). The biological function of CLEC3B has never been investigated in cholangiocarcinoma.

Here, we explored the clinical relevance of C-type lectin to cholangiocarcinoma. We also investigated the biological function of CLEC3B in bile duct cancer cells by overexpressing or knock-down the expression. The cancer inhibitory effect was investigated in vivo. The mechanism of CLEC3B inhibited cancer progression of cholangiocarcinoma was also explored.

Materials & Methods

Identification of the CLEC3B gene

The expression level of C-type lectin family genes was analyzed by using the TCGA database (GEPIA (Gene Expression Profiling Interactive Analysis) (cancer-pku.cn)), and the survival of differentially expressed genes was analyzed by Kaplan–Meier analysis with data from NODE database (Ling et al., 2023). The GEPIA visualization tool was used to screen the survival genes of CCA patients in the TCGA database according to the median expression level of each gene. The genes with survival significance in CCA patients were introduced into the CCA clinical cohort for survival analysis, and CLEC3B with survival significance was finally identified (Survival analysis of data from https://www.biosino.org/node/project/detail/OEP001105 statistics, deleted some incomplete data information).

CCA tissue samples and ethics statement

Thirty patients were diagnosed with CCA at the Affiliated Hospital of Qingdao University were included in the study, and the 30 samples included cancerous and para-cancerous tissues. The surgically resected tissues were cut to 0.2 cm*0.2 cm*0.2 cm and stored in a cryostorage tube containing RNA protective solution in a −80 °C refrigerator, or the tissues were cut into 1 cm3 and soaked in 4% paraformaldehyde solution and stored in a cool place at room temperature. This study was approved by the Ethics Committee of the Affiliated Hospital of Qingdao University, and all patients were fully informed and signed written informed consent (Ethics approval number: QYFY WZLL 28817).

Quantitative real-time PCR analysis

At room temperature, RNA was extracted from cells and tissues using Trizol solution, and the concentration was determined. The addition of 1 µg of template RNA and 4 times the concentration of gDNA wiper MIX to the Ep tube was carried out at 42 °C for 2 min to remove gDNA from the mixture. A total of 4 µL 5 ×HiScript III qRT SuperMix was added into the EP tube, mixed and centrifuged. The mixture was placed in a PCR instrument and the cDNA library was constructed under the following reaction conditions: 37 °C (15 min) → 85 °C (5 s) →4 °C(+ ∞). Dilute primer to 0.1OD for use (GAPDH pre-primer sequence: TGACTTCAACAGCGACACCCA, GAPDH post-primer sequence: CACCCTGTTGCTGTAGCCAAA, CLEC3B pre-primer sequence: AGCTCAAGAGCCGTCTGGACAC, CLEC3B post-primer sequence: GGAAGGTCTTCGTCTGGGTGAA). The premixed reaction system (19 µL/ well) and template cDNA (1 µL/well) were added into the eight-row reaction tube and detected by machine according to the reaction conditions provided in the instructions. The experiment made three secondary holes.

Cell lines and cell culture

Biliary duct cancer cell lines QBC939 and HUCCT1 were derived from the Shanghai Cell Bank of the Chinese Academy of Sciences. In brief, bile duct cancer cells were cultured in DMEM (Hyclone, Utah, USA) containing 10% fetal bovine serum and in a 5% CO2 incubator at 37 °C.

Cell transfection and lentiviral transduction

CLEC3B siRNA and plasmid were purchased from Jima (Shanghai, China) and Jikai (Shanghai, China), respectively. Sequence of CLECL3B-HOM-185: sense (5′–3′): AGAUGUUUGAGGAGCUCAATT, antisense (5′–3′): UUGAGCUCCUCAAACAUCUTT. CLEC3B siRNA was transfected into cells with Xfect™ RNA transfection reagent (Takara, Shiga, Japan). The plasmid was transfected with Lipofectamine 2000 (Invitrogen, Waltham, MA, USA). When the cell density is 60–80%, the siRNA or plasmid is transferred into the cell. QBC939 and HUCCT1 cells were infected with concentrated virus and cultured in complete culture medium containing serum and basic antibiotics for 24 h, and then screened with purinomycin to form stable cell lines. Overexpression of CLEC3B protocol: Cells in 10 cm dishes were cultured to 60%–70% and transfection began. 21 ug negative control plasmid (DNA),42 ul P3000 (transfection reagent) and 500ul serum-free high glucose medium (DMEM) were added to tube A (ep tube), 21 ug CLEC3B plasmid, 42 ul P3000 and 500 ul DMEM were added to tube B. Then 65 ul Lif3000 (transfer agent) and 500 ul DMEM were added to the two ep tubes respectively. The reagents were gently mixed and left for 10 min. The plasmids mixed with transfection reagents were added into cell culture dishes for 24 h–48 h to complete transfection.

Cell viability and colony formation assay

Cell viability was determined using CCK-8 kit (Dojindo Labs, Kumamoto, Japan). Cells were transfected with CLEC3B siRNA or plasmid and inoculated into 96-well plates 24 h later. After cell adhesion, cell activity at different time points was detected using Live Cell Counting Kit 8 (CCK8). Each well was added with 10 µL CCK-8 reagent and 100 ul DMEM and incubated at 37 °C and 5%CO2 for 2 h. Then absorbance optical density (OD) was measured at 450nm using a multifunctional enzyme marker. The transfected cells were inoculated on a 6-well plate (1,000 cells/well) and cultured for 14days for colony formation test. The cells were first fixed with paraformaldehyde, then stained with crystal violet, and finally, the number of cell colonies in the 6-well plate was counted to determine colony formation capacity.

Wound healing assays

The six-well cell culture plate was used for cell inoculation and cultured at 37 °C, 5% CO2. Transient transfection was performed after 24 h, and then cultured for 24 h. The cells were gently rinsed with sterile phosphate-buffered saline (PBS) to remove impurities such as cell debris and cultured with pure DMEM without fetal bovine serum. Then use the tip of a 200 µL straw to make a cross scratch at the bottom of the hole. Use a microscope (Nikon, Toyko, Japan) to capture images after 0 h and 24 h. ImageJ software (NIH, Bethesda, MD, USA) was used to measure the wound healing area and calculate the activity.

Invasion and migration assays

The invasion experiment was conducted in the 24-well Millicell chamber. Bile duct cancer cells were transfected and cultured at 37 °C for 24 h. Cells were suspended with DMEM of fetal bovine serum. After microscopic counting, cells with 200 µL serum-free medium (1.0 × 105) were added to the upper chamber (cell), and 600 µL DMEM containing 10% fetal bovine serum was added to the lower chamber (bottom of the pore plate) as a chemical attractant. The chamber was placed in the plate hole and the cell penetration was observed with a microscope. Remove the chamber and soak in methanol for 10 min to fix. The methanol is then poured out and soaked in 0.1% crystal violet solution for 30 min to stain. Finally, the cells were washed with water, the remaining crystal violet on the intima was wiped with a cotton swab, and the migrating cells were quantified at three random fields. The migration experiment was similar to the above, but without the Matrigel coating.

Vertebrate animal study methods

A total of 10 nude mice were purchased from Jinan Pengyue Experimental Animal Breeding Co., LTD. The strain is BALB/C NUDE, SPF grade. These animals were kept in the animal room of the Medical Animal Laboratory Center of the Affiliated Hospital of Qingdao University. Each cage has an independent ventilation system to ensure that no other bacteria interfere with the growth environment of nude mice. The nude mice were anesthetized with lidocaine, and then the nude mice were given subcutaneous injection of bile duct cancer cells (control group and overexpressed CLEC3B group) to make tumors. The size of the tumor was measured after it formed. After deep anesthesia, we de-necked the nude mice and killed them. There were no surviving animals at the end of the experiment.

Xenograft tumor formation model

The QBC939 stable transmutation cells overexpressing CLEC3B (1 × 106) and control QBC939 cells (1 × 106) were injected under the skin of 10 4-week-old male BALB/c thymus free nude mice (purchased from Jinan Pengyue Experimental Animal Breeding Co., LTD.). Within the next month, the nude mice were kept in a disease-free environment and the tumor volume was measured (the calculation formula: Volume (mm3) = length* width* height). After 30 days, the tumor was removed, the final volume was calculated, and the tumor tissue was preserved. This experiment was approved by the Animal Ethics Committee of the Affiliated Hospital of Qingdao University, and all experiments on nude mice were conducted in the pathogen-free medical animal laboratory of the Affiliated Hospital of Qingdao University (Ethics review number: AHQU-MAL20210210).

Western blotting

Total proteins were extracted from transfected cells and tissues in a ratio of 100:1:1 using a lysate consisting of RIPA buffer, phenylmethylsulfonyl fluoride (PMSF) and protease inhibitor mixture (PIC). The Enhanced BCA Protein Assay Kit (Beyotime Biotechnology, Shanghai, China) was used for quantitative analysis. An equal amount of total protein (30 µg) was electrophoretically transferred to a polyvinylidene fluoride (PVDF) membrane activated with methanol (0.45 µm; Millipore, Burlington, MA, USA). The protein was stained with labeled antibody (mouse or rabbit antibody) and the protein bands were detected by chemiluminescence kit (Affinity, San Francisco, CA, USA).

Statistical analysis

Statistical analysis was carried out with R language and GraphPad. The t test was used to compare the two sets of numerical continuous variables. Chi-square test was carried out for categorical variables. Log-rank checks survival analysis. P < 0.05 was statistically significant.

Results

The CLEC3B was decreased in cholangiocarcinoma

To study the role and mechanism of C-type lectin family proteins in cholangiocarcinoma, we analyzed the expression of the C-type lectin family genes in the TCGA database. Eight of the 12 genes, CLEC1B, CLEC2B, CLEC2D, CLEC3B, CLEC4G, CLEC4M, CLEC7A and CLEC11A, were differentially expressed in cholangiocarcinoma. Four of them were decreased in tumor tissues (Fig. 1A). Then Kaplan–Meier analysis was performed to screened out the genes related to overall survival (OS) in the NODE database. Only the CLEC2B, CLEC3B and CLEC11A showed statistical significance with survival rate. Overall survival was most strongly associated with high expression of CLEC3B, CLEC3b was selected for further study to explore the potential inhibitors that could be used as therapeutic target (Fig. 1B). The expression of cle3b protein was verified in four pairs of fresh tissues, and it was confirmed that the expression of CLEC3B was decreased in the tissues of cholangiocarcinoma (Fig. 1C).

Figure 1 Correlation between CLEC3B and patient prognosis in cholangiocarcinoma (CCA).

(A) The relative expression of partial of C-type lectin superfamily genes in cholangiocarcinoma in the TCGA database. (B) Kaplan–Meier analysis of cholangiocarcinoma prognosis in CCA database. The CLEC2B, CLEC3B and CLEC11A showed significant correlation with patient prognosis. (C) The protein expression of CLEC3B in four pairs of fresh cholangiocarcinoma tissues detected by western blot. (D) The mRNA expression of CLEC3B in 30 pairs of paracancer bile duct tissues and CCA tumor tissues was analyzed by RT-qPCR. (E) High expression of CLELC3B is associated with better survival in our patient’s cohort. The statistically significant differences are presented as: **p < 0.01; Student’s t-test.

The expression of CLEC3B mRNA in 30 pairs of cholangiocarcinoma tissues was analyzed in CCA tumor tissue and para-cancerous bile duct tissues. The expression of CLEC3B mRNA in tumor tissue significantly decreased in paracancer bile duct tissue (Fig. 1D). According to Kaplan–Meier data analysis, high expression of CLEC3B is associated with better patient outcomes (Fig. 1E). According to the analysis of clinicopathological factors, the expression level of CLEC3B was closely related to TNM stage and lymph node metastasis (Table 1).

Table 1 Correlation of CLEC3B expression with clinical and pathological factors.

	CLEC3B low (n = 15)	CLEC3B high (n = 15)	P value	
Age (year)			0.143	
>60	10	6		
≤60	5	9		
Gender			0.269	
Male	10	7		
Female	5	8		
CA19-9 (ng/ml)			0.195	
>39	13	10		
≤39	2	5		
CEA (ng/ml)			0.712	
>3.4	7	6		
≤3.4	8	9		
TNM staging			0.025	
I–II	3	9		
III–IV	12	6		
Lymph node metastasis			0.046	
Positive	7	2		
Negative	8	13		

CLEC3B inhibits the proliferation of CCA cells in vitro

The QBC939 and HUCCT1 cell lines were overexpressed, or knockdown with CLEC3B plasmid or si-RNA, respectively. The expression efficiency was verified by western blot. The overexpression of CLEC3B could significantly inhibit the proliferation of bile duct cancer cells as indicated by CCK-8 assay and colony formation. The knockdown of CLEC3B resulted in promotion of cellular proliferation (Figs. 2A–2D). The Cyclin D1 and C-myc were decreased with overexpressing CLEC3B, and up-regulated knockdown of CLEC3B (Fig. 2E). Moreover, after regulating the expression of CLEC3B, the BAX and Bcl-2 expression was also changed (Fig. 2F), suggesting that CLEC3B inhibits cellular proliferation and promoted cellular apoptosis.

Figure 2 Effect of CLEC3B on cell proliferation of bile duct cancer cells, QBC939 and HUCCT1.

(A–D) The expression of CLEC3B was verified in QBC939 and HUCCT1 cells after overexpression or knock-down. CCK8 assay was used to evaluate the cellular proliferation. (E–F) The c-Myc, Cyclin D1, BAX and Bcl-2 expression were detected. The statistically significant differences are presented as: *p < 0.05; **p < 0.01; ***p < 0.001; ****p < 0.0001; Student’s t-test.

The CLEC3B inhibits cell migration and invasion of bile duct cancer

To assess the effect of CLEC3B on the ability of cancer cells to migrate, wound healing and transwell experiments were performed. The cell mobility rate of overexpressed CLEC3B cells was significantly lower than that of control cells, while it was increased while knocking down the CLEC3B. The ability of overexpression of CLEC3B to significantly inhibit cell migration and invasion has been confirmed in transwell experiment, while knocking down CLEC3B can enhance the migration and invasion of cancer cells. These results suggest that the migration and invasion of bile duct cancer cells are negatively regulated by CLEC3B (Figs. 3A–3D). CLEC3B increased the expression of E-cadherin and decreased the expression of N-cadherin. The results show that CLEC3B affects EMT process (Fig. 3E).

Figure 3 Effect of CLEC3B on CCA cell migration and invasion.

(A–D) Transwell assay and wound healing assay were performed with QBC939 and HUCCT cells after regulating CLEC3B expression. (E) The effects of CLEC3B on the expression of EMT-related proteins E-cadherin and N-cadherin were detected by western blot. The statistically significant differences are presented as: *p < 0.05; **p < 0.01; ***p < 0.001; ****p < 0.0001; Student’s t-test.

The Ca2+ promotes the biological function of CLEC3B

The CLEC3B as a member of the C-type lectin superfamily, is a kind of transmembrane Ca2+ binding protein, we then confirmed whether Ca2+ affects the biological function of CLEC3B. Therefore, CaCl2 was added to complete culture medium of QBC939 and HUCCT1 that overexpressed CLEC3B to a final concentration of 0.8mmol/L. The proliferation ability of bile duct cancer cells was further inhibited after adding Ca 2+on the basis of overexpression of CLEC3B. The results of clonal formation were consistent with the above experiments. Transwell’s migration experiments showed that the migration ability was more significantly inhibited after Ca2+ addition. The CLEC3B may exert its biological effects depending on the presence of Ca2+ (Figs. 4A–4D).

Figure 4 Effects of Ca2+ on biological effects of CLEC3B.

(A–D) The effects of Ca2+ on bile duct cancer cells were detected by CCK8 method, clonal formation, transwell assay and wound healing assay after overexpressing CLEC3B. The statistically significant differences are presented as: **p < 0.01; ***p < 0.001; ****p < 0.0001; Student’s t-test.

CLEC3B regulates cell proliferation and metastasis through Wnt/β-Catenin signaling pathway

To reveal the CLEC3B-mediated gene regulatory network of CCA, we used RNA sequencing to analyze the transcriptional difference of QBC939 cells after CLEC3B overexpression. Compared with the control cells, the expression of 279 genes was up-regulated and 153 genes were down-regulated (p < 0.01) (Fig. 5A). KEGG analysis showed that signal transduction and tumor progression were significantly associated with these genes, including Wnt signaling pathway (Fig. 5B). After overexpressing CLEC3B in QBC939 and HUCCT1 cells, p-GSK3 β and β-catenin were decreased. These results indicated that, CLEC3B may regulates cell function through the Wnt/ β-catenin signaling pathway (Fig. 5B).

Figure 5 CLEC3B inhibit cellular function through Wnt/β-catenin signaling pathway.

(A) The RNA sequencing and KEGG analysis of QBC939 cell after CLEC3B overexpression. (B) The GSK3-β, p-GSK3-β, and β-catenin protein expression were analyzed. The statistically significant differences are presented as: *p < 0.05; **p < 0.01; ***p < 0.001; ****p < 0.0001; Student’s t-test.

The CLEC3B inhibits tumor formation in vivo

The role of cle3b in vivo was studied by subcutaneous xenograft model. The QBC939 overexpressing CLEC3B were used to construct an overexpression stable cell line. The CLEC3B overexpression stable cell line was inoculated subcutaneously of nude mice. The tumor volume was significantly smaller in the CLEC3B overexpressing group, indicating that CLEC3B can significantly inhibit the tumorigenesis of CCA cells (Fig. 6A). Immunohistochemical staining indicated Ki-67 and N-cadherin were lower in CLEC3B overexpressing group, while E-cadherin was higher in the control group (Fig. 6B). The protein expression was validated by western blot (Fig. 6C). It suggests that CLEC3B inhibit the tumor formation in vivo.

Figure 6 Effect of CLEC3B on tumor formation of bile duct cancer cells in vivo.

(A) The QBC939 cells stably overexpressed CLEC3B were injected subcutaneously. The tumor size was measured after one month. (B) Immunohistochemical staining were performed with formed tumor tissue. Ki-67, E-cadherin and N-cadherin were stained. (C) The protein expression of cellular proliferation, apoptosis, EMT and Wnt/β-catenin were analyzed with collected tumor tissues. The statistically significant differences are presented as: *p < 0.05; **p < 0.01; ****p < 0.0001; Student’s t-test.

Discussion

Cholangiocarcinoma is the second most common primary liver tumor in the world after hepatocellular carcinoma, and its morbidity and mortality remain high. Here, we investigate the biological effects of CLEC3B in CCA. The decreased CLEC3B expression in CCA was associated with poor survival. The inhibitory effect of CLEC3B on the progression of CCA was demonstrated both in vitro and in vivo. The calcium ions promoted the biological function of CLEC3B. The CLEC3B may work through Wnt/β-catenin signaling pathway.

The anticancer effects of lectins and the potential of lectins for cancer therapy have been demonstrated in previous studies (Yau et al., 2015). Here in this study, we deplored the C-type lectins family genes which related with the CCA survival. The high expression of CLEC1B, CLEC3B and CLEC11A are closely related to the good survival of cholangiocarcinoma. And we proved the anticancer effect of CLEC3B in CCA. In cervical cancer, longer survival was more strongly associated with higher expression of CLEC3B (Zhou et al., 2020). In lung cancer, CLEC3B expression is also reduced, and low expression of CLEC3B is an independent risk factor for disease-free survival (Sun et al., 2020). In clear cell renal cell carcinoma, CLEC3B also showed a significant downregulation trend (Liu et al., 2018). In accordance with previous studies, we found that CLEC3B could be a prognostic marker of CCA.

The mechanism of CLEC3B regulating CCA progression was also investigated. In CCA, the CLEC3B inhibited EMT process as indicated by E-cadherin and N-cadherin expression. The high expression of CLEC3B can also induce EMT in lung adenocarcinoma, and overexpressing CLEC3B enhanced the cell adhesion and limit the progression of cancer (Lu et al., 2022). In clear cell renal cell carcinoma, CLEC3B inhibited tumor growth through mitogen-activated protein kinase pathway. Overexpression of CLEC3B significantly reduced p38, P-p38, ERK and P-ERK. CLEC3B reduced the cell proliferation and differentiation, apoptosis and carcinogenesis (Liu et al., 2018). Another study found that CLEC3B, modified by H3K27ac, is involved in the PI3K-Akt signaling pathway in gastric function and tissue development of ruminant tumors (Kang et al., 2023). The molecular mechanism of CLEC3B still needs further investigation.

As the largest and most diverse family of mammalian carbohydrate-binding proteins, the common structure of C-type lectins is the Ca2+ dependent carbohydrate recognition domain. Among C-type lectins superfamily members, CLEC3B is closely related to human life development and cell aging (Keller & Rademacher, 2020). The C-type lectin receptor (CLR) recognizes carbohydrates by Ca2+ ions as cofactors and only a small percentage of CLRS recognize their ligands in the absence of Ca2+ (Drickamer & Taylor, 2015; Taylor & Drickamer, 2014; Weis, Drickamer & Hendrickson, 1992). Studies on the biological role of Ca2+ on CLEC3B gene are few. We confirmed that calcium ions can indeed promote the biological effects of CLEC3B in cholangiocarcinoma.

There are some limitations of the study. The multicenter and large case patient cohort is necessary to confirm the CLEC3B work as a prognostic marker. The biological function of CLEC3B depends on the presence of Ca2+, so its function may be affected by Ca2+ concentration. The mechanism of Ca2+ regulate CLEC3B needs future studies, and in vivo study may be necessary. The small molecule inhibitor arrived from CLEC3B need to further investigate the potential anticancer effect. More study needs to be done to explore the potential effects of CLEC3B.

Conclusions

In summary, CLEC3B plays a certain role in inhibiting the cancer progression in patients with CCA. The CLEC3B may inhibits cellular proliferation, migration and invasion through Wnt/β-catenin signaling pathway. CLEC3B could be a predictive biomarker for CCA patients’ prognosis and a potential therapeutic target of CCA.

Supplemental Information

Supplemental Information 1 30 Tumor tissue PCR and generation analysis

30 Original PCR data of human tumor tissue were generated to generate original data and original statistical map for analysis.

Supplemental Information 2 CCK8 assay of QBC939 cells, NC OE raw data, and original statistical map

Supplemental Information 3 QBC939 CCK8 NC SI raw data

CCK8 assay of QBC939 cells, NC OE raw data, and original statistical map.

Supplemental Information 4 Western blot raw data

Western blot original strip, quantitative gray value and statistical map.

Supplemental Information 5 Four pairs of fresh human tissue, verified by western blot

Supplemental Information 6 The survival analyses of genes in Fig. 1B and other genes not included in Fig. 1 and their raw data

Supplemental Information 7 TCGA raw data and images in Fig. 1A

Supplemental Information 8 The expression effect of clec3b transfected was verified

The original strips of western blot that validated the transfection efficiency of CLEC3B (Fig.2).

Supplemental Information 9 Raw tumor data of nude mice

The raw images of tumor size in nude mice, western blot strips of tumor tissue, and corresponding statistical maps.

Supplemental Information 10 HUCCT1 and QBC939 cck8 nc oe +Ca2+ raw images and data

Supplemental Information 11 Original pictures and data of functional experiments of HUCCT1 (Fig. 4) (Ca2+)

Supplemental Information 12 Original pictures and data of functional experiments of QBC939 (Fig. 4) (Ca2+)

Supplemental Information 13 Original pictures of HUCCT1 functional experiments (common overexpression group and control group, common knockdown rent and control group) as well as raw data and statistical charts

Supplemental Information 14 Original pictures of QBC939 functional experiments (common overexpression group and control group, common knockdown rent and control group) as well as raw data and statistical charts

Supplemental Information 15 MIQE checklist

Supplemental Information 16 Hightlight

Additional Information and Declarations

Competing Interests

Author Contributions

Human Ethics

Animal Ethics

Data Availability

The authors declare there are no competing interests.

Shengze Wu conceived and designed the experiments, performed the experiments, analyzed the data, prepared figures and/or tables, authored or reviewed drafts of the article, writing–original draft, Formal analysis, and approved the final draft.

Guangfeng Wang conceived and designed the experiments, performed the experiments, prepared figures and/or tables, make changes to the article, and approved the final draft.

Yuwei Xie analyzed the data, prepared figures and/or tables, tabular analysis, and approved the final draft.

Tingting Wu analyzed the data, prepared figures and/or tables, animal experimental design, and approved the final draft.

Fangqian Du performed the experiments, authored or reviewed drafts of the article, writing–review,editing, Formal analysis, and approved the final draft.

Chen Jin performed the experiments, authored or reviewed drafts of the article, writing–original draft, Data curation, and approved the final draft.

Bingzi Dong conceived and designed the experiments, authored or reviewed drafts of the article, formal analysis, and approved the final draft.

Chengzhan Zhu conceived and designed the experiments, analyzed the data, authored or reviewed drafts of the article, funding acquisition, Supervision, Validation, and approved the final draft.

The following information was supplied relating to ethical approvals (i.e., approving body and any reference numbers):

The Affiliated Hospital of Qingdao University approved research in its facilities (Ethical Application Ref: QYFY WZLL 28817)

The following information was supplied relating to ethical approvals (i.e., approving body and any reference numbers):

All animal experiments were conducted in the pathogen-free medical animal laboratory of the Affiliated Hospital of Qingdao University and were approved by the Animal Ethics Committee of the Affiliated Hospital of Qingdao University (Ethics review number: AHQU-MAL20210210)

The following information was supplied regarding data availability:

The raw measurements are available in the Supplementary File.

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
