# Peer review of "The CLEC3B inhibits cellular proliferation and metastasis of cholangiocarcinoma through Wnt/β-catenin pathway"

_PeerJ, doi:10.7717/peerj.18497_

## Round 0.1 · original submission · Major Revisions

I agree with the two reviewers' opinions. Please revise the paper according to their opinions. In particular, please provide more details about the experimental conditions and methods.

Reviewer 1 ·

Basic reporting

In the manuscript entitled “The mechanism of CLEC3B controlling proliferation and metastasis of cholangiocarcinoma through Wnt/β-catenin pathway”, the authors investigate the role of CLEC3B in cholangiocarcinoma, specifically how it influences cell proliferation, metastasis, and invasion through the Wnt/β-catenin signaling pathway. The experiments based on in vitro, in vivo, and in silico were performed in a technically sound manner. However, before publication, the authors must address the following several major points to improve the manuscript quality.

Experimental design

- The authors should provide detailed information on the technique used for the overexpression of CLEC3B in the Materials and Methods section.

Validity of the findings

- In Fig 1, Why did the authors not include CLEC2D and CLEC11A for further experiments, given that these genes also show differential expression between CCA and normal tissues?
- The authors should provide detailed information about the dataset used in Figures 1B and 1C.
- How do the authors explain the contradictory findings on the correlation between the expression of CLEC3B as shown in Figures 1B and 1C? The authors should address this in the discussion section.
- The authors should provide western blot images, band intensity quantification, and statistical testing to demonstrate the efficiency of the overexpression and knockdown techniques shown in Figure 2.
- In Figure 2, the authors performed western blot analysis on the expression of apoptosis-related molecules. However, the authors need to demonstrate whether the effects of CLEC3B overexpression and knockdown are due to cell cycle arrest or apoptosis.
- Could the authors provide band intensity quantification and statistical testing for all western blot experiments throughout the manuscript?
- To prove the effect of CLEC3B on the Wnt/β-Catenin signaling pathway in CCA xenograft mice, the authors should perform western blot, immunohistochemistry, or PCR to investigate the expression levels of molecules in the Wnt/β-Catenin signaling pathway in tissues from CCA xenograft mice.

Additional comments

-

Reviewer 2 ·

Basic reporting

1) There are some grammar errors in the paper, please carefully check and revise them.
2) The capitalization of genes and proteins is not standardized, it is recommended to check and correct it.

Experimental design

1) Fig 1F: There is no detailed follow-up information. It is recommended to supplement it.
2) There are only two groups for in vivo validation in Fig 4, and there are no NC and si-CLEC3B groups.
3) As a ca2+ binding transmembrane protein, this group was only set up in Fig 3J, and it is recommended to supplement this group in other experiments as well.

Validity of the findings

1) The standard deviation of Fig 1E is even greater than the mean, and the author needs to consider duplication and confirmation.
2) All clone formation assay, wound healing assay, migration assay, western blot assay, tumor volume, western blot assay requires quantitative statistical charts.
3) The data source for Fig 1c is unclear. In line 261 “by analyzing the data sets of published studies, it is found that CLEC1B and CLEC3B have survival differences in the survival correlation analysis of intrahepatic cholangiocarcinomas (FIG.1C).” But there is no reference.

Additional comments

no comment

---

## Round 0.2 · accepted · Accept

The reviewers are satisfied with your revisions. I am happy to accept your revised manuscript.

Reviewer 1 ·

Basic reporting

I am satisfied with the revision. Accept.

Experimental design

I am satisfied with the revision. Accept.

Validity of the findings

I am satisfied with the revision. Accept.